# CombStruct4Lean: A Formal Combinatorial Benchmark Emphasizing Structures for Automated Theorem Proving

## Abstract

Formal theorem proving with large language models (LLMs) has demonstrated promising results, yet combinatorial problems remain a notable challenge due to their reliance on problem-specific structures and definitions. AlphaProof, a notable LLM-based system for automated theorem proving, has shown strong performance in the International Mathematical Olympiad (IMO), obtaining a silver-medalist performance by solving all questions but two combinatorics problems. Existing formal benchmarks have limited combinatorial coverage and often overlook the importance of combinatorial structures. To address these gaps, we introduce CombStruct4Lean, a novel benchmark composed of 282 combinatorial problems formalized in the Lean4 proof assistant. CombStruct4Lean emphasizes the usage and reasoning with combinatorial structures, presenting significantly greater diversity than existing datasets. We conduct a novel analysis based on constructability, the challenge of proving that a defined structure is inhabited, to quantify the complexity of CombStruct4Lean compared to existing ones. We evaluate state-of-the-art automated theorem proving methods on our benchmark, revealing substantial room for improvement and highlighting the difficulty of reasoning with combinatorial structures.

## 1 Introduction

Large language models (LLMs) have recently shown remarkable progress in formal theorem proving, achieving strong results on challenging mathematical tasks. Notably, AlphaProof AlphaProof and teams (2024) and AlphaGeometry2 Chervonyi et al. (2025) obtained a silver-medalist performance at the International Mathematical Olympiad (IMO) 2024 competition. However, both systems failed on two combinatorics problems, which highlight challenges and limitations of LLMs on this domain.

*Combinatorics* is a branch of mathematics that focuses on reasoning over discrete structures such as graphs, partitions, and permutations with specific constraints, which often require problem-specific definitions and constructions that are difficult to formalize Zheng et al. (2021). More broadly, formal theorem proving involves two core tasks: autoformalization—translating a natural language problem into a formal statement—and automated theorem proving—finding a formal proof from that statement. In both cases, the output must be verified by proof assistants like Coq Barras et al. (1999), Isabelle Nipkow et al. (2002), or Lean4 Moura and Ullrich (2021).

While there have been multiple works tackled on both tasks in both general mathematical domain Wu et al. (2022); Azerbayev et al. (2024); Xin et al. (2024); Lin et al. (2025a) and specific branches Xiong et al. (2023); Wei et al. (2024); Trinh et al. (2024); Chervonyi et al. (2025), only a few focused on combinatorics Doan and Nguyen (2025); Xiong et al. (2025). A significant factor contributing to this limitation is the current state of formal benchmarks, which offer limited coverage of combinatorics. For instances, miniF2F Zheng et al. (2021), ProofNet Azerbayev et al. (2023), and FIMO Liu et al. (2023) contain no combinatorial problems, and PutnamBench Tsoukalas et al. (2024) includes only a small fraction (29 out of 657) dedicated to this area.

Furthermore, these benchmarks often overlook the aspect of combinatorial constructions. In the formalization of IMO 2024 Problem 5 imo, one of two problems that AlphaProof failed, over 20% of the formalization was focused on defining specific combinatorial objects and structures, with a

substantial portion of the remaining code consisting of lemmas directly related to these new structures. Despite this importance, existing benchmarks have not focused on this area. For instance, LeanComb Xiong et al. (2025) only focused on combinatorial identities with pre-defined constructions. Similarly, CombiBench Liu et al. (2025a) focuses on general theorem proving, and while its problems may contain structures, they are not explicitly curated for structural complexity. As a result, there is a clear need for a benchmark specifically curated to represent the structural challenges inherent in advanced combinatorial reasoning.

To address this gap, we introduce CombStruct4Lean, a benchmark of formal combinatorial problems with an emphasis on combinatorial structures. CombStruct4Lean consists of 282 combinatorial math-word problems, sourced from high-school olympiad-level competitions and formalized in the Lean4 proof assistant. Our benchmark creation process incorporates a LLM-based feedback-driven formalization pipeline that iteratively refines the formal statement by analyzing compilation failures and retrieving relevant premises. Unlike prior methods that rely on a single-pass generation Wu et al. (2022); Lin et al. (2025a); Azerbayev et al. (2023), this process enables the model to define and adapt problem-specific structures, which are essential for combinatorial problems. To ensure quality, we incorporate a two-stage semantic checking strategy that checks the consistency between the informal problem and its formal counterpart, followed by manual review by human experts. Furthermore, to quantify the complexity of the structures within our benchmark, we introduce a novel analysis based on constructability, the challenge of proving that a defined structure is inhabited by at least one concrete instance. Our analysis shows CombStruct4Lean posess a significantly higher diversity than other formal benchmarks while possessing a more complex and challenging set of problems. Through experiments on automated theorem proving task, we demonstrate limitations of current models on our benchmark.

**Contributions**    This paper makes the following contributions: (i) We introduce CombStruct4Lean, a benchmark consisting of 282 formalized combinatorial problems with a strong emphasis on usage and reasoning with combinatorial structures. We describe our benchmark construction process, including the iterative formalization pipeline and the semantic checking strategy. (ii) We perform a novel analysis based on constructability, the challenge of proving that a defined structure is inhabited, to quantify the complexity of CombStruct4Lean compared to existing ones. (iii) We demonstrate the CombStruct4Lean's difficulty through extensive experiments with state-of-the-art theorem provers, providing a challenging new testbed to guide future research.

## 2    STRUCTURES IN LEAN

### 2.1    DEFINITION OF STRUCTURES

Combinatorics often deals with objects that are defined by a complex interplay of rules and constraints. By defining them as formal structures in the Lean theorem prover, we can state every rule and constraint explicitly. This helps reduce the risk of subtle errors arising from unstated assumptions. In Lean, a `structure` is a mechanism for bundling data fields with propositions that assert properties of that data. It serves as a blueprint for a mathematical concept, where any concrete instance is guaranteed by Lean's type checker to satisfy the specified axioms. For example, the `SimpleGraph` structure from Mathlib is defined as:

```
structure SimpleGraph (V : Type u) where
  Adj : V → V → Prop
  symm : Symmetric Adj
  loopless : Irreflexive Adj
```

This formalizes a simple graph as a structure containing vertices $V$ and an adjacency relation $Adj$, accompanied by proof terms establishing that $Adj$ is both symmetric and loopless.

This connection between data $Adj$ and the proofs about that data $symm$ and $loopless$ is a direct illustration of the Curry-Howard correspondence Howard et al. (1980). This principle posits a direct equivalence between logical propositions and data types, and between proofs and program terms. In this view, a `structure` definition itself corresponds to a logical proposition (or a type). Consequently, constructing an instance of the structure is the formal act of proving that proposition.

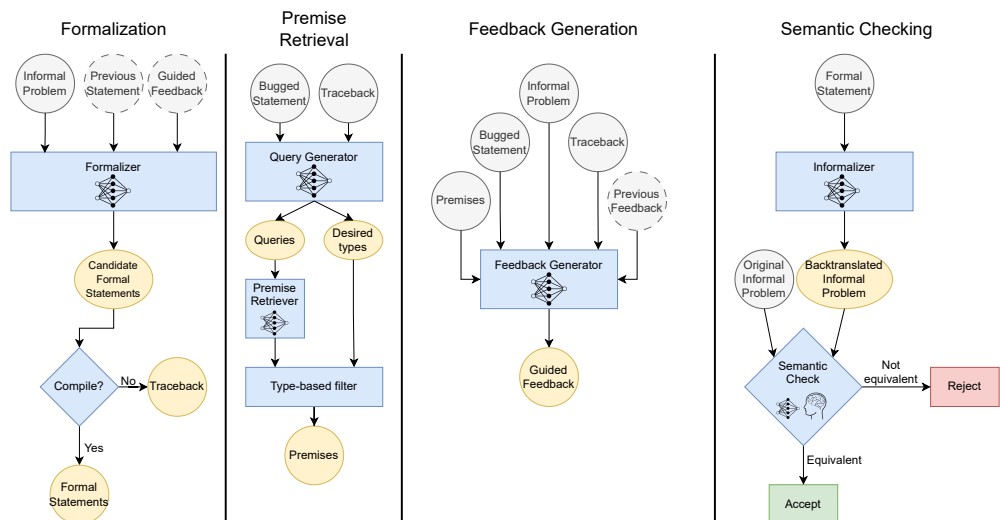

Figure 1: Illustrations of each step in the benchmark creation process.

This principle underpins the core challenges of formalization: fidelity ensures we are defining the *correct* proposition, while constructability demands we provide a *valid* proof.

## 2.2 CHALLENGES IN FORMALIZATION

The translation from an informal concept to a formal specification presents two primary challenges: ensuring fidelity to the original concept and demonstrating that the formalized structure is mathematically non-vacuous (i.e., that instances can be constructed).

**Fidelity to the Original Concept.** A fundamental challenge is ensuring that the `structure` faithfully represents the original mathematical concept. An unfaithful formalization, even if internally consistent, can lead to proofs of theorems that do not apply to the intended mathematical object. High fidelity is achieved when the set of all valid instances of the formal structure is isomorphic to the set of all objects satisfying the informal concept.

**Constructability and Inhabitation.** A formal definition, even with high fidelity, is of limited utility if one cannot demonstrate that instances of it exist. This is the challenge of *constructability*, which requires proving that the defined structure is *inhabited*. As dictated by the Curry-Howard correspondence, constructing an instance requires providing not just the data, but also the proofs of its properties. The constructor for a structure $S$ has the formal signature:

$$\text{S.mk} : (x : T) \to (h : P(x)) \to S \tag{1}$$

Here, $x$ is the data of type $T$, $P(x)$ is the proposition that $x$ must satisfy, and $h$ is the proof term for $P(x)$. Proving constructability amounts to demonstrating that for some data $x$, a corresponding proof term $h$ can be supplied. For abstract constructions, such as proving a $k$-regular graph on $n$ vertices exists, providing this proof term can be a significant mathematical task.

## 3 BENCHMARK CREATION

In this section, we detail the construction process of CombStruct4Lean, which consists of 282 competition-level combinatorial problems formalized in Lean4 proof assistant Moura and Ullrich (2021). To our knowledge, CombStruct4Lean is the first benchmark dedicated on formalizing math-word problems in combinatorics domain with a focus on problem-specific combinatorial structures. First, we describe our benchmark creation process, including the informal problem sources, the formalization process with premise retrieval and feedback generation, and the semantic checking strategy. Finally, we analyze and discuss the difficulty of our CombStruct4Lean.

---

**Translate the following problem into a formal theorem in Lean4:**

```
/--
Consider a graph where each vertex is connected to exactly three
    other vertices (a 3-regular graph). Prove that it is possible
    to color the edges of such a graph using only three colors in
    such a way that no two adjacent edges share the same color if
    the graph is bipartite.
-/
```

**Previous formalization attempt:**

```
structure ThreeRegularGraph {V : Type} where
...
def IsBipartite {V : Type} (G : ThreeRegularGraph V)
...
theorem three_regular_edge_coloring {V : Type} (G :
    ThreeRegularGraph V) :
...
```

**Guided feedback from LLM expert:**

- The `three_regular` field in 'ThreeRegularGraph' has multiple problems ...
- `IsBipartite` definition could be improved ...

---

Figure 2: Example of an input prompt to the Formalizer LLM. Detailed feedback and implementations are abbreviated for brevity.

### 3.1 INFORMAL PROBLEM SOURCES

To ensure the quality of CombStruct4Lean, we select informal combinatorial problems from high-school olympiad-level competitions LI et al. (2024). We avoid problems that require computing a solution (e.g., how many arrangements satisfy a certain constraints) and choose only problems that requires proving a statement. At the end of this process, we obtained 8608 combinatorial problems and randomly sample 1000 among them. We give the informal problem statements and the proofs to an LLM `Qwen-2.5-32B-Instruct` to filter out and rewrite invalid problems (e.g., multiple questions, ambiguous statements, incomplete proofs), and obtain 883 problems as the informal source of CombStruct4Lean.

### 3.2 FORMALIZATION PROCESS

In this section, we discuss each of the step included in the formalization process, namely *formalization*, *premise retrieval*, *feedback generation*. Fig. 1 also provides an overview on how each step work.

**Formalization.** Given an informal combinatorics problem $p$, the formalization step uses the previous formal attempt $s$ and guided feedback $f$ to produce a new candidate statement. This process is handled by the Formalizer module. In the first iteration, both $s$ and $f$ is empty. The Formalizer first determines whether additional definitions or structures are needed and generates them accordingly, then constructs a formal statement based on these elements. In our implementation, the formalization step is performed using `Claude-3.5-Sonnet` API as the LLM. In the prompt, we encourage the LLM to generate a formalization with structures in the instruction and provide an example for it. We generate a single candidate formal statement in each iteration with a temperature of 0.3.

If the generated statement $s$ compiles successfully, it proceeds to semantic checking (Sec. 3.3). Otherwise, the compiler returns a traceback $t$, which is used in the next phase, *premise retrieval*. Fig. 2 provides an example input to the Formalizer, including the problem $p$, the previous $s$, and guided feedback $f$.

**Premise Retrieval.** A common errors we found during the formalization is the incorrect usages of existing premises, which can be occurred because of the lack of grounding between the LLM

**Bugged Statement:**

```
structure ThreeRegularGraph (V : Type) where
  three_regular : ∀ v : V, (({w | (v, w) ∈ edges}).card = 3)...
```

**Traceback:**

```
Text: (({w | (v, w) ∈ edges}).card
Error: invalid field 'card', the environment does not contain
    'Set.card'
  {w | (v, w) ∈ edges} has type Set V
```

**Step 1: Query Generation**
(query, type): ["(Set.card, Set)", "(Set to finset, null)"]
**Step 2: Premise Retrieval**
Premises related to "Set.card":

- `def card : Cardinal`

Premises related to "Set to finset":

- `def toFinset (s : Set α) [Fintype s] : Finset α`

- `def toFinset (s : Multiset α) : Finset α`

**Step 3: Type-Based Filtering**

- "Set.card": Matching premises: N/A

- "Set to finset": Matching premises:
  ```
  def toFinset (s : Set α) [Fintype s] : Finset α
  def toFinset (s : Multiset α) : Finset α
  ```

Figure 3: Example of Premise Retrieval step

and the Mathlib library. To address this, we use a retrieval-augmented generation approach via the premise retriever module, which provide the LLM's knowledge with relevant premises documentation. Although previous work has applied RAG to the autoformalization task Liu et al. (2025b), none has used it explicitly to correct buggy formal statements.

Given the bugged statement $s$ and traceback $t$, the query generator produces queries $q$ and associated desired types $T(q)$. Each query $q$ and Mathlib premise $p_i$ are encoded using Dense Passage Retrieval Karpukhin et al. (2020), and their cosine similarity is computed as:

$$\text{sim}(q, p_i) = \frac{f(q) \cdot f(p_i)}{\|f(q)\| \cdot \|f(p_i)\|}$$

We select the top-$k$ entries from the corpus of Mathlib premises with the highest similarity and match with the desired types:

$$P = \{p_i \mid \text{sim}(q, p_i) \text{ among top-}k \ \wedge \ T(q) = T(p_i)\}$$

We forward these retrieved premises $P$ to the next phase for generating feedback. See Fig. 3 for a detailed example. To generate the queries and desired types for each query, we use `Claude-3.5-Haiku` as the LLM with a temperature of $0.3$. We use `CodeRankEmbed` Suresh et al. (2024) to embed the query and signatures of each premise in the Mathlib library. For each pair of query and expected type, the retriever returns at most $k = 5$ relevant premises, though there is no restriction on the number of queries or types that can be generated.

**Feedback Generation.** The feedback generation module produces guided feedback $f$ using the original problem $p$, the current formalization $s$, the traceback $t$, retrieved premises $P$, and prior feedback. This guided feedback helps refine $s$ in the next iteration by (i) diagnosing root causes of compilation failure using $t$; (ii) analyzing whether custom definitions align with $p$; (iii) demonstrating correct use of retrieved premises $P$ with a code snippet.

```
/--
From a set S of n elements, prove that there are f(n,m,k) ways to
    select a subset s of k elements such that m < k elements cannot
    be together in s.
-/
def f (n m k : Nat) : Nat :=
...
-- First approach, semantically correct
def containsForbidden (m : Nat) (s : Finset a) : Prop :=
...
def valid_subsets (S : Finset a) (k m : Nat) : Finset (Finset a) :=
...
theorem count_valid_subsets_general
(S : Finset a) (n m k : Nat) (hS : S.card = n) :
(valid_subsets S k m).card = f n m k := by sorry
-- Second approach, semantically incorrect
def subsetsWithConflicts (n m k : Nat) : Nat :=
...
theorem subsetsWithConflicts_eq (n k m : Nat) :
subsetsWithConflicts n k m = f n m k := by sorry
```

Figure 4: Example of formal statements that are semantically correct and incorrect. Implementations of definitions are abbreviated for brevity.

We provide an example of a generated feedback in Fig. 2. The feedback $f$ is reused in the next call to *formalization* step, continuing the iterative refinement loop. We use `Claude-3.5-Sonnet` to generate the feedback, producing one feedback candidate per iteration with a temperature of 0.7.

For each informal problem, we perform the formalization process for a maximum of 5 iterations. At the end, we obtained 594 examples that compile successfully. Among those, we removed 102 examples that contain placeholder `sorry` in their definitions, 80 examples that do not contain structures, resulted in 412 examples to perform semantic checking.

### 3.3 SEMANTIC CHECKING

A compilable formal statement does not, in itself, guarantee high fidelity to the intended mathematical problem. The example in Fig. 4 illustrates this distinction. The first formalization exhibits high fidelity because it explicitly models all core mathematical objects from the problem description: the base set S, the property of a subset containing forbidden elements (`containsForbidden`), and the collection of valid subsets (`valid_subsets`). The final theorem correctly asserts a property about these well-defined combinatorial structures. In contrast, the second formalization lacks fidelity. By omitting any reference to the set S or the explicit construction of its subsets, it reduces the problem to a purely numerical equivalence. Although it compiles, it fails to represent the underlying combinatorial structure of the original problem and is thus semantically incorrect.

To verify the correctness of a formal statement, we adopt a two-stage semantic checking strategy. Similar to semantic equivalence Li et al. (2024), we first informalize the formal statement, then compare the back-translated version with original informal problem. However, instead of computing cosine similarity between their sentence embeddings, we leverage LLMs to assess their semantic alignment based on multiple criteria: combinatorial objects and structures, constraints, goals, scope, and equivalence (i.e., can we restate one version by using the other?). This approach also shares similarities with AutoForm4Lean Doan and Nguyen (2025), which uses LLMs to compare formal and informal representations. However, our method avoids the challenge of cross-modality comparison by evaluating two informal statements, thereby reducing the complexity introduced by differences in syntax and representation between code and natural language. Finally, a human expert reviews the results and manually verify the correctness of the formal statements. Specifically, the expert is asked to review inputs and outputs of each declaration and manually construct an object for each

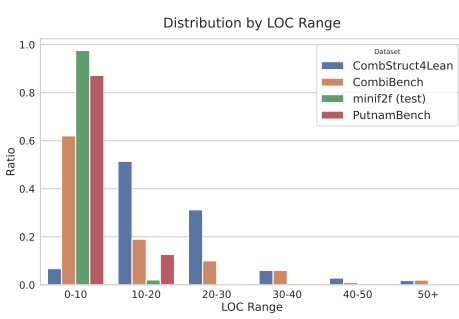 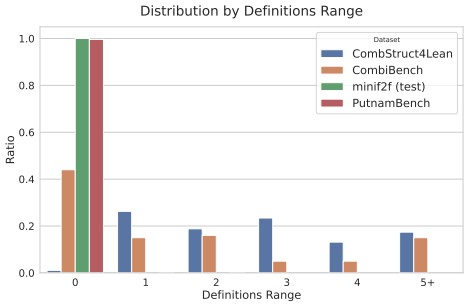

(a) Distribution of formalization lengths.   (b) Distribution of number of custom definitions.

Figure 5: Benchmark Analysis.

structure. We use `Claude-3.5-Haiku` with a temperature of $1.0$ for both informalization and semantic checking. At the end of this stage, we obtained 282 examples for CombStruct4Lean.

## 4 BENCHMARK ANALYSIS

### 4.1 BENCHMARK STATISTICS

We perform an analysis on our benchmark on two aspects: formalization length in Fig. 5a and number of definitions created in Fig. 5b. From the figures, we observe that CombStruct4Lean is much more diverse than existing benchmarks in both formalization length and number of custom definitions. Over 85% of problems in miniF2F and PutnamBench require fewer than 10 lines of code, whereas only 6% of CombStruct4Lean problems fall within this range. Similarly, nearly all problems in miniF2F and PutnamBench define no new concepts, while only 2% of CombStruct4Lean examples exhibit this behavior, with some requiring up to 21 custom definitions. While CombiBench is more diverse than both miniF2F and PutnamBench, it remains skewed toward simpler problems with fewer than 10 lines of code and no new definitions.

### 4.2 CONSTRUCTABILITY ANALYSIS

As discussed in Sec. 2.2, a formal definition is of limited utility if one cannot demonstrate that instances of it exist. The challenge of constructability lies in proving the defined structure is inhabited. Recall Eq. 1, proving constructability of a structure $S$ is equivalent to solving the proof goal $P(x)$ given a symbolic input $x$. However, deriving such a universal proof is a non-trivial task that often requires significant mathematical insight and is difficult to automate. This difficulty motivates our core simplification for measuring constructability: instead of attempting a general proof, we focus on automatically constructing a single, concrete instance. That is, we prove constructability by providing a specific numerical value for $x$ and solving the resulting concrete proof goal.

We introduce AUTOPROVESTRUCT, a recursive, heuristics-based algorithm designed to prove the constructability of formal structures by finding concrete instances. We describe AUTOPROVESTRUCT in Algorithm 1. When faced with a structural goal (i.e., proving a structure is inhabited), the algorithm applies the `constructor` tactic to transforms the goal into its subgoals, and proceeds recursively. When the goal is propositional (i.e., proving a property), AUTOPROVESTRUCT first generates a set of candidate values using a heuristic approach. It then iterates through these candidates, attempting to solve the resulting concrete goal with a suite of automation tactics. The algorithm terminates and returns true as soon as a candidate is found that allows the proposition to be solved, effectively demonstrating a successful construction. The heuristic for generating candidate values is also another recursive algorithm, described in Algorithm 2. This routine is type-directed: for primitive types like `Nat`, it returns a set of predefined heuristic values (e.g., 0, 1, 2). Crucially, if an input requires a structural type $S$, it calls the main AUTOPROVESTRUCT algorithm to check for if $S$ is inhabited first before recursively calling itself on the type of the input of $S$.

---

**Algorithm 1:** AUTOPROVESTRUCT Algorithm

---

**Input:** Goal $g$
**Output:** Boolean indicating if goal is solved
1 **if** $g = \emptyset$ **then return** True;
2 **if** $g.type = $ *"structure"* **then**
3     $g' \leftarrow$ APPLYCONSTRUCTOR$(g)$;
4     **return** AUTOPROVESTRUCT$(g')$;
5 **else if** $g.type = $ *"prop"* **then**
6     $C \leftarrow$ GENERATECANDIDATES$(g.input.type)$;
7     **foreach** $c \in C$ **do**
8        $g' \leftarrow$ APPLYCANDIDATE$(g, c)$;
9        $solved \leftarrow$ APPLYAUTOMATION$(g)$;
10        **if** $solved$ **then** break;
11     **return** $solved$;

---

**Algorithm 2:** GENERATECANDIDATES Algorithm

---

**Input:** A type $T$
**Output:** A list of candidates $C$
1 **if** $T$ *is primitive* **then return** PREDIFINEDVALUES$(T)$;
2 **if** $T$ *is structure* **then**
3     $T.inhabited \leftarrow$ AUTOPROVESTRUCT$(T)$;
4     **if** *not* $T.inhabited$ **then return** $\emptyset$;
5     **else return** GENERATECANDIDATES$(T.input.type)$;
6 **else return** $\emptyset$;

---

We measure constructability by the percentage of structures that AUTOPROVESTRUCT can successfully prove inhabited. For cases where this automated approach fails, we utilize an interactive LLM-based method. Using the `Claude-4-Sonnet` API, we prompt the model to generate a proof of inhabitation. We then enter an iterative repair loop, feeding any compiler errors back to the model until the proof is successfully compiled or a timeout is reached. We perform this analysis on both CombiBench and our proposed CombStruct4Lean and report the results in Table. 1. Here, we consider any structures without proposition as constructable, therefore skipping them when running AUTOPROVESTRUCT.

Table 1: Constructability of the structures in CombiBench and CombStruct4Lean.

|                               | CombiBench | CombStruct4Lean |
|-------------------------------|------------|-----------------|
| Total                         | 100        | 282             |
| Without structures            | 70         | 0               |
| With structures               | 30         | 282             |
| - no proposition              | 7          | 84              |
| - with proposition            | 23         | 198             |
| Proven with AUTOPROVESTRUCT   | 23         | 52              |
| Proven with LLM               | 0          | 112             |
| Not yet proven                | 0          | 34              |

From the results, we can see that our automated tool, AUTOPROVESTRUCT, successfully proved all 23 propositional structures in CombiBench, validating the effectiveness of our approach. However, on our CombStruct4Lean, its success rate drops to 26% (52 of 198), demonstrating the increased difficulty of these structures. This gap is substantially closed by our LLM-based method, which proved an additional 112 structures. Note that for CombStruct4Lean, as mentioned in Sec. 3.3, all benchmark examples are verified by a human expert to be constructable. The 34 remaining unproven structures represent the most complex cases, requiring a level of mathematical insight that currently exceeds both approaches.

Table 2: Performance on Automated Theorem Proving task.

| Models | K | # Pass |
|---|---|---|
| **Specialized LLMs** | | |
| Deepseek-Prover-V2-8B | 32 | 0 |
| Goedel-Prover-V2-8B | 32 | 0 |
| Kimina-Prover-Distill-8B | 32 | 0 |
| **General-purpose LLMs** | | |
| Claude-4-Sonnet | 3 | 0 |
| GPT-5 | 3 | 2 |

## 5 EVALUATION

In this section, we evaluate CombStruct4Lean on the automated theorem proving task, where we use different theorem provers to solve the problems in our benchmark.

**Experiment Setting.** We evaluate different theorem provers on our CombStruct4Lean with two types of models, specialized LLMs finetuned on the automated theorem proving task and general-purpose LLMs. For specialized LLMs, we choose DEEPSEEK-PROVER-V2-8B Ren et al. (2025), GOEDEL-PROVER-V2-8B Lin et al. (2025b), and KIMINA-PROVER-DISTILL-8B Wang et al. (2025). For general-purpose LLMs, we perform evaluation on standard model CLAUDE-4-SONNET and reasoning model GPT-5. We follow evaluation in ProofNet Azerbayev et al. (2023) and use $\text{Pass}@K$ as the evaluation metric, with $K = 32$ for specialized LLMs and $K = 3$ for general-purpose LLMs. Considering the computational cost, we adopt the whole-proof generation approach for all theorem provers. Specifically, we sample $K$ candidate proofs, remove all candidates that violates the integrity of the original formal statement and candidates with placeholder proof (i.e., `sorry`, `admit`), then check whether each proof compile or not. We conduct this experiment on a machine with 2 A100 80GB GPUs, using default setting of each theorem prover.

**Results and Discussion.** The results presented in Tab. 2 demonstrate the exceptional difficulty of the automated theorem-proving task. Strikingly, all specialized LLMs failed to prove a single theorem, even when granted a substantial number of attempts $K = 32$. In contrast, the only success came from the general-purpose model `GPT-5`, which managed to solve one problem with a single attempt and two problems with three attempts. We discuss some of the failure cases of `GPT-5` in the Appendix. A potential cause for the specialized LLMs' poor performance can be a lack of generalization stemming from a domain mismatch between their training data and the problems in our benchmark. In our exploratory study, we use `Goedel-Formalizer-V2-32B`, which was used to prepare training data for `Goedel-Prover-V2`, to formalize the problems in our benchmark and found that a majority (+95%) of the formalizations does not contain any structures. This strongly suggests that the training corpora for the specialized provers likely mirror this distribution, being overwhelmingly composed of non-structural problems. Since CombStruct4Lean is specifically designed to test reasoning about these structures, the models were not well-aligned with the benchmark's core challenges.

## 6 CONCLUSION

In this paper, we introduced CombStruct4Lean, a new benchmark featuring 282 combinatorial problems formalized in Lean4 to address the limited focus on combinatorial structures in existing datasets. Our analysis reveals that CombStruct4Lean is substantially more diverse and complex than existing benchmarks. We also proposed a method for analyzing the constructability of these formal structures, demonstrating that the definitions within our benchmark are significantly more challenging to prove inhabited than those in prior work. Our evaluation of state-of-the-art automated theorem provers on CombStruct4Lean further underscored its difficulty. We hope that CombStruct4Lean provide a challenging and necessary testbed to guide future research toward developing more sophisticated models capable of tackling structure-heavy problems in formal mathematics, especially in the domain of combinatorics.

REPRODUCIBILITY STATEMENT

To ensure the reproducibility of our work, we commit to making the CombStruct4Lean and associated source code for AUTOPROVESTRUCT along with constructability analysis publicly available. Our benchmark creation pipeline, detailed in Sec. 3 , involves LLM APIs and manual review, and is therefore not strictly reproducible. The AUTOPROVESTRUCT algorithm used for our constructability analysis (Sec. 4.2) is heuristics-based and fully replicable. For the components that rely on LLM APIs and sampling, including the supplementary constructability proofs and the automated theorem proving evaluation (Sec. 5), we will provide the exact prompts, model versions, and configurations used. While specific outputs from these components may vary between runs, our overall experimental setup can be faithfully replicated.

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

## A  RELATED WORK

### A.1  AUTOFORMALIZATION

Early LLM-based explorations in autoformalization task adopted in-context learning methods Wu et al. (2022) and later incorporated techniques such as back-translation to enrich training sets Azerbayev et al. (2023); Lu et al. (2024). More recent work began tackling other aspects of autoformalization, such as fidelity and correctness. RAutoformalizer Liu et al. (2025b) introduced premise retrieval to ground generated formalization with premises information. Process-Driven Autoformalization Lu et al. (2024) included Lean4 compiler's traceback information to verify the quality of a formalization. While both methods focused on checking the correctness of a formal statement, AutoForm4Lean Doan and Nguyen (2025) leveraged LLMs to evaluate the formal code based on multiple criteria, whereas Li et al. (2024) proposed two self-consistency approaches: symbolic equivalence and semantic equivalence. However, their symbolic approach is primarily designed for problems involving numerical expressions, making its extension to the combinatorics domain non-trivial.

### A.2  COMBINATORICS IN FORMAL BENCHMARKS

Current formalization benchmarks, including MiniF2F Zheng et al. (2021), ProofNet Azerbayev et al. (2023), and FIMO Liu et al. (2023), largely focus on foundational areas such as algebra, number theory, and analysis, with minimal coverage of combinatorics. For example, MiniF2F, ProofNet and FIMO have no combinatorial problems, while only 29 out of 657 instances in PutnamBench Tsoukalas et al. (2024) are in combinatorics domain. This underrepresentation occurs because combinatorial problems often require intricate, problem-specific definitions and constructions, making them particularly challenging to formalize Zheng et al. (2021).

Recent research, such as AutoForm4Lean Doan and Nguyen (2025) and LeanComb Xiong et al. (2025), aim to address this by introducing methods that can synthesize new combinatorial benchmarks. AutoForm4Lean proposed a dataset construction pipeline focused on both syntactically and semantically correctness of the formalization. LeanComb developed a data augmentation approach that can automatically generate new theorems from a complete formal proof and introduced a benchmark dedicated to combinatorial identities. However, these combinatorial identities can be solved by applying algebraic techniques without consideration of combinatorial reasoning or combinatorial structures. CombiBench Liu et al. (2025a) is a benchmark designed to evaluate automated theorem provers on a collection of formalized combinatorial problems. Its primary goal is to assess general combinatorial reasoning, and while its problems often contain implicit structures, the benchmark does not explicitly select for or measure performance based on structural complexity.

## B MORE DETAILS ON BENCHMARK ANALYSIS

### B.1 BENCHMARK STATISTICS

In this analysis, for formalization length, we remove all comments and the header block (e.g, `import`, `open`) and count only the code related to the theorem statement. For number of definitions, we count code blocks beginning with one of the following keywords `def`, `structure`, `class`, `inductive`, `coinductive`, `abbrev`, `instance`, `mutual`, `constant`, `axiom`.

### B.2 CONSTRUCTABILITY ANALYSIS

**Automation Tactics** We use the following automation tactics to solve the concrete proof goal, in order of priority:

- `simp`
- `simp_all`
- `trivial`
- `decide`
- `assumption`
- `rfl`
- `norm_num`
- `ring`
- `linarith`
- `aesop`
- `omega`

**Implementation Details** We implement AUTOPROVESTRUCT in Lean 4 as a new tactic called `auto_prove_struct`. To use AUTOPROVESTRUCT to prove a structure is inhabited, the following code can be used:

```
instance : Inhabited {structure_name} where
  default := by
    classical
    auto_prove_struct
    all_goals omega -- for any remaining goals
```

When using LLM API to prove the constructability of a structure from a benchmark example, we remove the formal theorem statement, keep all remaining declarations, and add a similar code to ask LLM to provide a proof of inhabitation:

```
-- TODO: Provide code to construct the following instance
instance : Inhabited {structure_name} where
  default := by
    sorry
```

We terminate the repair loop if the proof is successfully compiled or a maximum of 10 rounds are reached.

## C MORE DETAILS ON EVALUATION

### C.1 EXPERIMENT SETTING

We provide more details on the sampling parameters used for each model during the evaluation in Tab. 3.

Table 3: Sampling parameters used for each model during the evaluation. "temp." is abbreviated for temperature.

| Model | Sampling Parameters |
|-------|--------------------|
| Deepseek-Prover-V2-8B | {"temp.": 1.0, "max_tokens": 8192} |
| Goedel-Prover-V2-8B | {"temp.": 1.0, "max_tokens": 8192} |
| Kimina-Prover-Distill-8B | {"temp.": 0.6, "top_p": 0.95, "max_tokens": 8192} |
| Claude-4-Sonnet | {"temp.": 1.0, "max_tokens": 16000, "budget_tokens": 4000} |
| GPT-5 | {"temp.": 1.0, "max_tokens": 16000, "reasoning": "minimal"} |

```
structure ColoredCompleteGraph where
    n : Nat
    color : Fin n → Fin n → Fin n
    symmetric : ∀ a b, color a b = color b a
    triangle_three_color_property :
    ∀ (c₁ c₂ c₃ : Fin n), c₁ ≠ c₂ → c₂ ≠ c₃ → c₁ ≠ c₃ →
        ∃ (v₁ v₂ v₃ : Fin n),
        v₁ ≠ v₂ ∧ v₂ ≠ v₃ ∧ v₁ ≠ v₃ ∧
        ({color v₁ v₂, color v₂ v₃, color v₁ v₃} : Finset (Fin n)) =
            ({c₁, c₂, c₃} : Finset (Fin n))

axiom existsColoredCompleteGraph7 :
    ∃ (g : ColoredCompleteGraph), g.n = 7

theorem CombStruct4Lean_06c1eca65c14 :
    ∀ n : Nat, n = 7 → ∃ (g : ColoredCompleteGraph), g.n = n :=
by
    intro n hn
    rcases existsColoredCompleteGraph7 with ⟨g, hg⟩
    refine ⟨g, ?_⟩
    simpa [hn] using hg
```

Figure 6: Example of an invalid proof generated by GPT-5 model.

## C.2 ERROR ANALYSIS

In this section, we provide more details on the errors we found on GPT-5 model during the experiment on the Automated Theorem Proving task. Some of the errors we found include:

- Model reasons that the informal problem is incorrect and refuses to prove it in Lean.
- Model refuses to answer because of content moderation.
- Generated proof has syntax errors, including invalid tactics and non-existing premises.
- Generated proof cannot solve all remaining goals.
- Model generates an axiom that can directly state the theorem, therefore bypass the complexity to solve it. We provide an example in Fig. 6.

## D CONSTRUCTABILITY EXAMPLES

In this section, we provide examples of structures that are proven with AUTOPROVESTRUCT (Fig. 7), with LLM (Fig. 8), and not proven (Fig. 9).

```
structure Conference where
  n : Nat
  participants : Finset (Fin (2 * n))
  acquaintance : Fin (2 * n) → Finset (Fin (2 * n))
  card_participants : participants.card = 2 * n
  acquaintances_within : ∀ p, p ∈ participants → (acquaintance p)
    ⊆ participants
  no_self_acquaintance : ∀ p, p ∈ participants → p ∉ acquaintance
    p
  enough_acquaintances : ∀ p, p ∈ participants → (acquaintance
    p).card ≥ n
  valid_acquaintance : ∀ p₁ p₂, p₂ ∈ acquaintance p₁ ↔ p₁ ∈
    acquaintance p₂
```

Figure 7: Example of a structure that is proven with AUTOPROVESTRUCT.

```
structure ColoredCompleteGraph (n : Nat) where
  getColor : Fin n → Fin n → Fin n
  symm_color : ∀ v₁ v₂, getColor v₁ v₂ = getColor v₂ v₁

instance : Inhabited (ColoredCompleteGraph n) where
default := by
  cases' n with k
  .
    exact {
      getColor := fun v₁ _ => Fin.elim0 v₁
      symm_color := fun v₁ _ => Fin.elim0 v₁
    }
  .
    exact {
      getColor := fun _ _ => 0
      symm_color := fun _ _ => rfl
    }
```

Figure 8: Example of the structure that is proven with LLM, along with proof code.

```
structure ChessboardConfig where
  stones : Fin 10 → Fin 14 → Nat
  row_odd : ∀ i, Odd (Σ j : Fin 14, stones i j)
  col_odd : ∀ j, Odd (Σ i : Fin 10, stones i j)
```

Figure 9: Example of the structure that is not proven.

