# OpenReview forum: "A Formal Combinatorial Benchmark Emphasizing Structures for Automated Theorem Proving"
_ICLR.cc/2026/Conference — ICLR 2026 Conference Withdrawn Submission_

### Official Review · Reviewer_HdZ5 · 2025-10-28

**Soundness:** 3
**Presentation:** 3
**Contribution:** 2
**Rating:** 4
**Confidence:** 3

**Summary:**

This paper introduces CombStruct4Lean, a new benchmark designed to address the challenges that LLMs face in formal theorem proving for combinatorics. The authors point out that existing benchmarks—such as miniF2F and PutnamBench—offer limited coverage of combinatorial mathematics and overlook the importance of newly defined structures in combinatorics. Even advanced systems like AlphaProof have failed on combinatorial problems from the IMO. CombStruct4Lean consists of 282 combinatorial problems formalized in Lean4, each containing a problem-specific structure. In addition, the paper proposes a new algorithm AUTOPROVESTRUCT to prove the constructibility of these structures.

**Strengths:**

- This paper introduces CombStruct4Lean, a new benchmark focusing on combinatorics problems with problem-specific structures. It contains 282 high-school competition-level combinatorial problems, and each problem has a new defined structure in its formal statement.
- The paper proposes a new algorithm, AUTOPROVESTRUCT, designed to prove the constructibility of structures. It successfully demonstrates the constructibility of all structures in CombiBench and of 52 structures from CombStruct4Lean.

**Weaknesses:**

- The paper does not mention recent advances of IMO in ATP. For example, Seed Prover has successfully solved a combinatorics problem in IMO 2025. Current trends in this field are shifting toward higher-level mathematics (e.g., undergraduate, phd, and even open research problems). In contrast, this benchmark still focuses mainly on high-school-level combinatorial problems. Furthermore, in the domain of formal mathematics, other branches such as algebra and topology also involve rich problems that need to define some new structures. The authors currently focus only on combinatorics, making the scope limited.
- CombiBench points out that the difficulty of combinatorics lies in the need to introduce new definitions or structures. It already includes a number of combinatorial problems containing some new definitions or structures, allowing for a certain degree of evaluation of model performance in this domain. From Figure 5, however, CombStruct4Lean seems to be an extension of part of CombiBench, increasing the quantity and difficulty of the problems.

**Questions:**

- According to Table 2, most models score zero on the automated theorem proving task. How well do human experts perform on this benchmark? How to prove the data quality? Although the paper verifies the constructibility of structures, it does not deeply analyze the correctness of the formal statements, but only checks semantic consistency by human experts. Thus, it can not determine whether the low scores are caused by the wrong formalization or the wrong definitions/structures.
- In formalization tasks, an informal mathematical problem can often be formalized in many different ways, and the same applies to structure or definition. Based on past experience, definitions, formal statements, and formal proofs which are generated by LLM are usually longer and more complicated than those written by human experts. Since the structures and statements in CombStruct4Lean are primarily generated by LLMs and checked for semantic consistency by human experts, how do the authors prevent redundant, overly complex, and unnecessary formalization details to increase the problem’s difficulty?

---

### Official Review · Reviewer_cMbq · 2025-10-30

**Soundness:** 3
**Presentation:** 2
**Contribution:** 3
**Rating:** 6
**Confidence:** 2

**Summary:**

This paper introduces CombStruct4Lean, a mathematical benchmark focused on combinatorial problems formalized in Lean4. it is  motivated by the the observation that existing formal mathematics benchmarks lack tasks that involve rich combinatorial structures. CombStruct4Lean consists of 282 formalized problems curated through an LLM-assisted formalization pipeline, with compiler feedback, retrieval, and semantic validation steps. The authors further analyze their dataset’s properties (e.g., formalization length, number of custom definitions) and demonstrate that it is more complex than existing benchmarks. They also evaluate several theorem-proving LLMs, showing that they struggle on this benchmark.

**Strengths:**

- The motivation for introducing more complex combinatorial structures into mathematical benchmarks is important and well-founded.
- The fact that the authors formalize all questions into correct Lean4 makes the benchmark more accessible and reusable for future research.

**Weaknesses:**

- In Table 2, although the evaluation demonstrates poor performance of some provers and general-purpose models, the authors do not evaluate strong reasoning-optimized models (e.g., Gemini-2.5-pro) that currently lead on mathematical reasoning benchmarks. GPT-5 with “minimal” thinking effort (as specified in the Appendix) does not exhibit strong reasoning performance, so it is unsurprising that it performs poorly.
- Several key notions introduced as core contributions are insufficiently or ambiguously defined. For example, the notion of a structure being “inhabited” is used throughout but not clearly defined before “constructability” is introduced. As a result, the definition of constructability remains somewhat unclear. I also feel like "isomorphic" for characterizing "high fidelity" could be clearer defined. These definitions are central to the paper’s conceptual contribution and would benefit from clearer exposition.

**Questions:**

- It seems that the construction process removes more than 70% of the originally selected problems. Is this simply because LLM formalization outputs can be noisy, making it most efficient to discard everything that is semantically incorrect or fails to compile? If this formalization process were generalized to other mathematical domains, would we also expect around 70% of problems to be removed?
- Regarding the quantitative analysis: the number of definitions introduced and the length of the formalization do not necessarily reflect a problem’s combinatorial structural complexity. Do the authors also provide any qualitative analysis to complement these results?
- One small typo: line 204, “both s and f [are] empty.”

---

### Official Review · Reviewer_dta1 · 2025-10-31

**Soundness:** 1
**Presentation:** 2
**Contribution:** 1
**Rating:** 2
**Confidence:** 4

**Summary:**

This paper introduces CombStruct4Lean, a benchmark consisting of formalized combinatorics problems in Lean to evaluate automated theorem proving with LLMs. The authors construct this benchmark by taking informal statements from high-school competitions (NuminaMath-1.5) and formalizing them using a pipeline that consists of several stages: (1) formalization until it compiles using claude-3.5-sonnet, (2) semantic checks using LLMs and a human expert. The eventual benchmark consists of 282 samples. Results show that model significantly struggle with the benchmark, often obtaining 0%.

**Strengths:**

- The paper correctly identifies combinatorics as a category of tasks which is paid lesser attention to in Lean, trying to address it.
- The generation approach is somewhat interesting, containing non-trivial components that for instance use a variation of RAG to fix syntax errors.

**Weaknesses:**

This paper has one critical weakness and several other weaknesses. In particular, the data is of very low quality, which kills the benchmark. I have structured this section in three: one describing the critical weakness, one describing other weaknesses, and one describing smaller points that have not affected my judgment.

## Critical Weakness
**The data is of very poor quality. In short, the data contains contradictory samples, duplicates, and was not rigorously enough verified.** Throughout the paper, there are various markers that made me strongly suspect this and that need clarification (see later), but 5 minutes of looking at the data gives a much clearer and worrisome picture. For instance, the following four samples are four different informal statements from the benchmark:
```
On a board there are $n$ nails each two connected by a string. Each string is colored in one of $n$ given distinct colors. For each three distinct colors, there exist three nails connected with strings in these three colors. Prove that $n$ can be 6.

$n$ nails are connected pairwise by strings. Each string is colored in one of $n$ given colors. For any three distinct colors, there exist three distinct nails whose three pairwise strings use exactly those three colors (one per edge). Prove that $n$ cannot be 6.

On a board there are $n$ nails each two connected by a string. Each string is colored in one of $n$ given distinct colors. For each three distinct colors, there exist three nails connected with strings in these three colors. Prove that $n$ cannot be 7.

$n$ nails nailed on a board are connected by two via a string. Each string is colored in one of $n$ given colors. For any three colors there exist three nails connected by two with strings in these three colors. Prove that $n$ can be 7.
```
Not only are these statements very similar, they are **contradictory**. It cannot both be the case $n$ can and cannot be 6 or 7. I found this issue within five minutes of looking at the dataset, clearly indicating gross negligence, and making me doubt whether any human expert has looked at this dataset thoroughly (see later). Furthermore, the dataset contains other (near)-duplicates. For instance:

```
In a volleyball tournament, 110 teams participated, each playing exactly one game against each of the others (there are no ties in volleyball). It turned out that in any group of 55 teams, there is one team that lost to no more than four of the other 54 teams in this group. Prove that in the entire tournament, there is a team that lost to no more than four of the other 109 teams.

In a volleyball tournament, 110 teams participated, each playing exactly one game with each of the others (there are no ties in volleyball). It turned out that in any group of 55 teams, there is one that lost to no more than four of the other 54 teams in this group. Prove that in the entire tournament, there is a team that lost to no more than four of the other 109 teams.
```
There are many more samples that are very similar in the dataset. This once again indicates negligence in the dataset construction process, and form a clear rejection reason for a paper whose main contribution is a benchmark. In particular, it indicates that the dataset does not contain 282 samples, but much less once deduplicated. Furthermore, the paper does not sufficiently address the following points:
- As rightly pointed out, formalizing combinatorics problems in Lean is very hard. Yet, the authors mainly rely on the use of an outdated model (Claude-3.5-Sonnet) to fully formalize each statement. In contrast, benchmarks like PutnamBench and CombiBench, which contain problems that are easier to formalize, were constructed using months of manual labor by Lean experts. It is never discussed how subtle mistakes by Claude-3.5-Sonnet could potentially fool the human expert.
- Furthermore, it is never thoroughly discussed how the human expert is used. In particular, the following questions need to be answered:
   - What are the qualifications of this expert?
   - How long did this expert spend on verifying each sample? Formalizing a single combinatorics problem properly can easily take 4 hours (likely even more), so even verifying a statement should take a lot of time. Any human spending more than five minutes on each sample should have seen the many duplicates and the contradictory statements. Because they did not, the process could not have been done rigorously.
   - How many samples did the human expert edit or discard? The low number of discarded samples in the semantics check (reduction by factor of 2, even though this is by far more difficult that syntax correctness) makes me worried that the pass was not done rigorously enough.
   - The **only** fully rigorous check whether a data sample is not pure noise, would be to check if it can be proven. The authors should do this with the human expert on a random subset of samples.
- The models obtain 0%: this is a very strong indication that something might be wrong. Rather than an indication of difficulty, it could be that it is an indication of noise. Yet, the authors never discuss this point.
- GPT-5 almost **never** says an input statement is wrong if it is not (due to sycophancy). Yet, the authors mention it is one of the main reasons that GPT-5 obtains a poor performance. This is very worrisome. The authors should check the reasons for GPT-5 saying the input statement is wrong and make sure it is not actually a problem in their data (which it for sure is in some cases, as shown above).

## Weaknesses
- CombiBench (a paper cited by the authors) is another benchmark for combinatorics in Lean. While the authors argue they are somewhat different,  these differences as currently presented are unconvincing and minor. The fact that CombiBench was constructed using manual labor to avoid noise is also a clear advantage of CombiBench.
- AutoProveStruct performs worse than calling an LLM in an iterative repair loop. Therefore, it does not outperform its own baseline. As such, the method not only distracts from the main contribution of the paper (the benchmark), it is a net negative contribution in itself. The authors should remove it from the paper.
- The authors should provide the code to the reviewers for review. In particular given the low quality of the data, it needs reviewing.


## Remarks
- Please use citet and citep appropriately. Not putting your citations in brackets makes things very hard to read (I continuously want to read the next word, only to find that it is actually a citation). As a rule of thumb: citations should be in brackets unless it forms an essential part of the sentence. For instance, "XXX et al. introduced a benchmark", but "A benchmark was previously introduced to address this issue (XXX et al.)".
- There are a few typos throughout the paper:
   - L44 is not a proper sentence. ("that" is missing)
   - L370 transforms -> transform
   - L466 reference the specific appendix instead of mentioning that it is in the appendix.
   - L315: "the" missing
   - L458: compile -> compiles
- Instead of referencing AlphaProof, which is proprietary and not much is known about it, authors would do better to reference newer systems like SeedProver or Hilbert.
- The benchmark is not more diverse than prior benchmarks, it is more specific by targeting a subarea of mathematics. For instance, PutnamBench tackles a wider range of mathematics. The authors redefine what diversity means in section 4.1 by saying it is based on the number of lines in the formalization, but this indicates difficulty (or noise), not diversity.

**Questions:**

see above.

---

### Official Review · Reviewer_eUUV · 2025-11-01

**Soundness:** 2
**Presentation:** 2
**Contribution:** 2
**Rating:** 4
**Confidence:** 3

**Summary:**

This paper introduces CombStruct4Lean, a benchmark consisting of 282 combinatorial problems designed to evaluate LLMs’ ability to write Lean programs that solve proof problems. Unlike prior works, it focuses on combinatorial structures. The paper presents an LLM-based, feedback-driven pipeline that transforms informal problem descriptions into formal Lean statements, and proposes an algorithm to estimate constructability as a measure of problem difficulty. Empirical results show that the benchmark is challenging, with both specialized and frontier reasoning models solving few, if any, of the problems.

**Strengths:**

- The motivation for testing LLMs’ ability on combinatorial structures is solid, given it is often overlooked in prior works.
- The paper isolates autoformalization from automated theorem proving. Its main innovation lies in constructing an LLM-driven pipeline for autoformalization, and evaluating model behaviour solely on automated theorem proving.
- The proposed AUTOPROVESTRUCT algorithm to measure constructability and quantify problem difficulty is novel.

**Weaknesses:**

- The LLM-driven pipeline for formalizing problem statements is a multi-round process with several stages in each round, involving multiple LLMs (from different models and even model families). This complexity introduces risks of inconsistency and hallucination. Moreover, many stages of the pipeline build upon ideas from prior works, albeit with adaptation and integration, which limits novelty.
- While the benchmark is difficult, it offers limited insights into how specialized or frontier reasoning models perform, as most fail entirely. A wider range of problem difficulties, with easier ones, would allow more fine-grained and informative analysis, helping future research on model improvement.
- The evaluation is limited to five models, with little discussion or analysis of results, including the error analysis in Appendix on GPT-5. Given that this is a benchmark paper, a more extensive and insightful evaluation would better demonstrate its value.
- Although autoformalization is treated as part of preprocessing, an ablation study where LLMs are given informal problem statements and asked to produce Lean solutions could strengthen the paper and clarify why separating these two processes is important.

**Questions:**

- What is the rationale for using different LLMs (and model families) at various stages of the pipeline?
- What does the “sorry” placeholder refer to (Line 300)?

---

### Official Review · Reviewer_y8jF · 2025-11-05

**Soundness:** 3
**Presentation:** 2
**Contribution:** 3
**Rating:** 2
**Confidence:** 4

**Summary:**

This paper introduced CombStruct4Lean for combinatorial problems. CombStruct4Lean includes 282 competition-level combinatorial problems formalized in Lean proof assistant. Questions are drawn from high-school olympiad-level competitions of NuminaMath-1.5. Compared to the previous dataset with combinatorial problems of CombiBench, the proposed CombiBench has several benefits: the size, formal structure annotation of Lean. They also introduced a recursive, heuristics-based algorithm for formal proving of AUTOPROVESTRUCT. The difficulty of the proposed dataset of CombStruct4Lean is also suggested as many more problems are unsolvable even with either LLMs or AUTOPROVESTRUCT.

In abstract, they state that they evaluate state-of-the-art auto- mated theorem proving methods on our benchmark, revealing substantial room for improvement and highlighting the difficulty of reasoning with combinatorial structures. However, the experiments are quite limited to three existing models and single size of 7B. The domain is obviously limited to combinatorial structures. Hence, I am not sure this study surely matches the wide interests of the community.

**Strengths:**

- S1: The dataset proposal with combinatorial problems with formal structure of Lean.
- S2: CombStruct4Lean is surely larger than CombiBench and has questions that are always formalized in Lean as of Table1.
- S3: They also introduced a recursive, heuristics-based algorithm for formal proving of AUTOPROVESTRUCT.

**Weaknesses:**

- W1: The scope is limited to combinatorial problems.
- W2: The dataset size of 282 is limited, although it is larger than the small CombiBench dataset. The relationship of this dataset compared to the existing famous large-scale dataset of NuminaMath-LEAN is not discussed.
- W3: The entire dataset is drawn from other existing datasets and hence some aspects, e.g., difficulty of it comes from the previous set.
- W4: Limited Table 2: both the number of the Specialized LLMs and General-purpose LLMs are limited or examined in the limited model-size / K-size. Indeed, “Proven with LLM” in Table 1 suggests that a number of problems in CombStruct4Lean are solved solely using LLMs while almost no problems are solved within Table 2.

**Questions:**

Q1: “Proven with LLM” in Table 1: Can you provide the details of this experiments? What kind of LLM did you use (Claude-4-Sonnet API ?) and in what settings did you obtain this result? How it is different from the setting of Table 2?
Q2: Can you also discuss the relationship between CombStruct4Lean and NuminaMath-LEAN?

---

### Note · Authors · 2025-11-14

I have read and agree with the venue's withdrawal policy on behalf of myself and my co-authors.